# Decentralized UAV Swarm Scheduling with Constrained Task Exploration Balance

**Runfeng Chen, Jie Li * and Ting Peng**

College of Intelligence Science and Technology, National University of Defense Technology, Changsha 410073, China
*   Correspondence: lijie09@nudt.edu.cn

**Abstract:** Scheduling is one of the key technologies used in unmanned aerial vehicle (UAV) swarms. Scheduling determines whether a task can be completed and when the task is complete. The distributed method is a fast way to realize swarm scheduling. It has no central node and UAVs can freely join or leave it, thus making it more robust and flexible. However, the two most representative methods, the Consensus-Based Bundle Algorithm (CBBA) and the Performance Impact (PI) algorithm, pursue the minimum cost impact of tasks, which have optimization limitations and are easily cause task conflicts. In this paper, a new concept called "task consideration" is proposed to quantify the impact of tasks on scheduling and the regression of the task itself, balancing the exploration of the UAV for the minimum-impact task and the regression of neighboring tasks to improve the optimization and convergence of scheduling. In addition, the conflict resolution rules are modified to fit the proposed method, and the exploration of tasks is increased by a new removal method to further improve the optimization. Finally, through extensive Monte Carlo experiments, compared with CBBA and PI, the proposed method is shown to perform better in terms of task allocation and total travel time, and with the increase in the number of average UAV tasks, the number of iterations is less and the convergence is faster.

**Keywords:** UAV planning; swarm scheduling; distributed method; market-based algorithm; task consideration

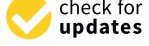



## 1. Introduction

At present, unmanned aerial vehicle (UAV) swarms have attracted increasing research attention because of their excellent reliability, rapidity, and self-organization. Compared with a single-UAV system, a UAV swarm can complete more tasks that cannot be completed by a single UAV, such as multi-UAV cooperative detection or transportation, which can be used in search and rescue, logistics distribution, environmental monitoring, and so on [1–3]. UAV scheduling is one of the key technologies of a swarm system, which determines whether the task can be completed and when the task is complete [4,5].

The most traditional way to complete UAV scheduling is the centralized method, which centralizes the information of all UAVs and tasks in one node, solves the optimal scheduling in this node, and then distributes the scheduling results to all UAVs. This method can achieve good optimization performance, but the solution time increases exponentially with the increase in the number of UAVs and tasks. Although there are heuristic methods such as the genetic algorithm [6,7], the ant colony algorithm [8,9], and particle swarm optimization [10,11] which search for feasible sub-optimal solutions or do not pursue optimal solutions using heuristic information, the computation time is still long if the swarm scale is large. The reason for this is that the essence of swarm scheduling is an NP-hard combinatorial optimization problem, similar to the multiple traveling salesman problem (MTSP) [12,13] but more complex, with constraints such as task deadlines. In addition, the centralized method also has a central node. If the central node is damaged, the UAV swarm will collapse, and there is a vulnerability problem. There are also studies

that turn a single center into multiple centers where, as long as the input information is consistent, the same scheduling output can be obtained through the same algorithm. This redundancy improves the robustness, but the calculation explosion problem still cannot be avoided [14,15].

A more advanced approach is the distributed method, which has no central node and allows UAVs to freely join or leave, thus being more robust and flexible. The market-based method is the most popular distributed method, which ensures no conflict between tasks in UAV scheduling in the swarm by means of independent selection and distributed negotiation. Each UAV independently selects its own suitable task and then communicates with the swarm to resolve the existing task conflicts. This does not require a consistent environment or other situational information obtained by each UAV, but only requires the consistency of the scheduling after negotiation, which greatly reduces the communication traffic and improves the robustness of the system [16,17]. In addition, since each UAV performs calculations independently, the amount of calculation is greatly reduced, and each UAV could further reduce the amount of calculation required by using heuristic methods, such as the greedy algorithm, so that the timeliness of the swarm scheduling is satisfied. Through several iterations of the 'scheduling calculation and conflict resolution' stages, swarm scheduling without task conflicts is obtained.

Research on market-based methods can be divided into two categories: one is basic scheduling and the other is scheduling extension, which is shown in Table 1. The most representative base scheduling algorithms are the Consensus-Based Bundle Algorithm (CBBA) [18] and the Performance Impact (PI) algorithm [19], where CBBA introduces a basic mechanism of two-stage iteration and PI introduces a novel concept to evaluate tasks. Scheduling extension is used to extend the basic scheduling, such as by rescheduling [20,21] to adapt to dynamic environments, probability-tuned scheduling [22,23] to improve robustness, and others [24,25] to enhance optimization. The method proposed in this paper belongs to basic scheduling, which proposes a new concept, task consideration, to further improve the performance of scheduling. At present, all these methods are still greatly affected by communication [26,27], and how to reduce the impact of the network on scheduling is also one of our future research directions, which is not involved in this paper.

**Table 1.** Table on classification and comparison of market-based methods.

| Categories | Methods | Contributions | Limitations |
|---|---|---|---|
| basic scheduling | CBBA [18] | basic mechanism | network |
| | PI [19] | novel concept | network |
| | the proposed method | novel concept | network |
| scheduling extension | rescheduling [20,21] | dynamic regulation | network |
| | probability-tuned [22,23] | robust performance | network |
| | others [24,25] | optimization | network |

Among these market-based methods to base scheduling, the PI algorithm [19,21,24] is the state-of-the-art method. It reduces the total travel time of global scheduling by minimizing the impact of tasks on UAV scheduling in real time. However, it blindly pursues the minimum cost impact and ignores the exploration of the adjacent tasks of the UAV, which can lead continuous outward exploration, resulting in the failure to complete the adjacent tasks of the UAV before the arrival of the deadline. In addition, outward exploration easily causes task conflicts with other UAVs, and task contention can only be solved by conflict resolution, resulting in more iterations. In order to enable the UAV swarm to complete more time-sensitive tasks and avoid task conflicts with other UAVs to reduce the number of iterations, this paper not only pays attention to the impact of tasks on scheduling but also pays attention to the regression of UAV scheduling, that is, UAVs also focus on the selection of adjacent tasks.

In this paper, a new concept is proposed, task consideration, which firstly defines and quantifies the impact of tasks on scheduling and the regression of the task itself,

balancing the exploration of the UAV for the minimum impact task and the regression of neighboring tasks to improve the optimization and convergence of scheduling. In addition, the conflict resolution rules are modified to adapt to the method proposed in this paper, and the exploration of tasks is increased by a new removal method to further improve the optimization of the method. Finally, through a large number of Monte Carlo experiments, compared with the two most representative algorithms, CBBA and PI, the number of tasks allocated by the proposed method is increased by 6% at most. Under the same number of allocated tasks, the number of samples that reduces the total travel time of UAV swarm scheduling accounts for 65–96% and 31–48% of the total, respectively, and the travel time is reduced by 2–6% and 1–4%, respectively. When the average number of tasks per UAV is high, the improved convergence speed of the proposed method is more obvious, which is much lower than the other two methods.

The rest of the paper is organized as follows. Section 2 introduces the basic symbols used in this paper and formulates the UAV swarm scheduling problem. Section 3 introduces the proposed method, where the concept of task consideration, the basic idea, and specific process are described. In Section 4, a large number of experiments are conducted to verify the validity of the proposed method. Finally, Section 5 concludes this paper.

## 2. Preliminaries

### 2.1. Symbol Definition

In order to make it easier to read and understand the proposed method, the basic symbol definitions and descriptions used in this paper are shown in Table 2.

**Table 2.** The basic symbol definitions and descriptions used in this paper.

| Symbol | Description |
|:---:|:---:|
| $n$ | number of UAVs |
| $m$ | number of tasks |
| $i, k$ | UAV ID |
| $j$ | task ID |
| $\mathbf{p}_i, \mathbf{p}_k$ | the schedule of UAV $i$ and UAV $k$ |
| $\mathcal{I}$ | the set of all UAV IDs |
| $\mathcal{J}$ | the set of all task IDs |
| $c_{ij}$ | the cost of UAV $i$ to perform task $j$ |
| $a_i$ | the maximum number of tasks that UAV $i$ is able to perform |
| $t_{ij}$ | the time of UAV $i$ to perform task $j$ |
| $d_j$ | the deadline of task $j$ |
| $\eta_{ij}$ | the task consideration of UAV $i$ to perform task $j$ |
| $c_i$ | the total cost of UAV $i$ |
| $c_i^o(\{j\})$ | the regression value of the task $j$ for UAV $i$ |
| $\eta_{ij}^*$ | the task consideration of UAV $i$ to add task $j$ |
| $y_{ij}$ | the winning bid of task $j$ considered by UAV $i$ |
| $z_{ij}$ | the winner of task $j$ considered by UAV $i$ |
| $\mathbf{p}_i \ominus \{j\}$ | the schedule of UAV $i$ after removing task $j$ |
| $\mathbf{p}_i \oplus \{j\}$ | the schedule of UAV $i$ after adding task $j$ |

### 2.2. Problem Formulation

UAV swarm scheduling is a planning process in which $n$ UAVs complete $m$ tasks sequentially. Each UAV $i$'s schedule $\mathbf{p}_i$ is the execution sequence and time of allocated tasks, satisfying task constraints such as starting time and deadlines. Generally, it could be formulated as a constrained optimization problem with objectives formulated as follows:

$$\min \sum_{i=1}^{n} \left( \sum_{j=1}^{m} c_{ij}(\mathbf{p}_i) \right) \tag{1}$$

subject to

$$|\mathbf{p}_i| \leq a_i \quad \forall i \in \mathcal{I} \tag{2}$$

$$\bigcup_{i=1}^{n} \mathbf{p}_i = \mathcal{J}, \quad \mathbf{p}_i \cap \mathbf{p}_k = \varnothing \quad \forall i, k \in \mathcal{I} \tag{3}$$

$$t_{ij}(\mathbf{p}_i) \leq d_j \quad \forall i \in \mathcal{I}, \quad \forall j \in \mathcal{J} \tag{4}$$

where $c_{ij}(\mathbf{p}_i)$ is the cost for UAV $i$ to perform task $j$ according to its schedule $\mathbf{p}_i$. The first constraint indicates that the number of tasks in UAV schedule $\mathbf{p}_i$ cannot exceed its capability $a_i$; the second indicates that the swarm should complete all tasks, and the tasks in UAV $i$'s schedule $\mathbf{p}_i$ and UAV $k$'s schedule $\mathbf{p}_k$ should not coincide; the third is that the time $t_{ij}$ of any task $j$ performed by any UAV $i$ according to its schedule $\mathbf{p}_i$ cannot exceed task $j$'s deadline $d_j$.

## 3. Method

### 3.1. Basic Idea

Here, a new concept, task consideration, is proposed to express the cost of tasks for UAV scheduling, where the impact of the task on scheduling and the regressivity of the task itself is quantified and first defined as follows:

$$\eta_{ij}(\mathbf{p}_i \ominus \{j\}) = c_i(\Delta \mathbf{p}_i) + c_i^o(\{j\}) \quad \forall j \in \mathcal{J}, \quad \forall i \in \mathcal{I} \tag{5}$$

where $\eta_{ij}(\mathbf{p}_i \ominus \{j\})$ is the removal consideration of task $j$ for UAV $i$ provided that task $j$ is removed from the scheduling $\mathbf{p}_i$. The term before the plus sign is the influence value of the task on the scheduling, which is obtained by the difference in scheduling cost with and without removing the task j, $c_i(\Delta \mathbf{p}_i) = c_i(\mathbf{p}_i) - c_i(\mathbf{p}_i \ominus \{j\})$. The term after the plus sign is the regression value of the task to the UAV, which is calculated by the UAV $i$'s location $\mathbf{l}_{io}$ and task $j$'s location $\mathbf{l}_{\{j\}}$, $c_i^o(\{j\}) = \left\| \mathbf{l}_{io} - \mathbf{l}_{\{j\}} \right\|_2$.

The task consideration balances the exploration and regression of UAVs on tasks well, so that the UAV does not constantly explore outward in the greedy pursuit of the minimum impact on scheduling, and it strengthens the UAV's exploration of adjacent tasks to moderately escape from the local optimum the greedy strategy becomes trapped in. This is not only helpful for the number of completed tasks, because the proximity exploration avoids the problem of ignoring the adjacent tasks in the continuous outward exploration, but it also avoids the possibility of task conflicts with other UAVs due to continuous outward exploration, thus reducing the number of iterations required for the method to resolve conflicts.

Figure 1 intuitively shows the difference between the proposed method and the traditional method, where the traditional method selects tasks only based on the impact of those tasks on the schedule, and the proposed method also considers the regression of tasks. For example, in the traditional method, when UAV-1 makes a decision to choose task $T_2$ or task $T_3$, it respectively calculates the total cost increase $c_{12}$ after task $T_2$ is added to its schedule, and the total cost increase $c_{13}$ after task $T_3$ is added to the schedule. IT then chooses the task $T_3$ with the minimum total cost increase, which will cause a task conflict with UAV-2. However, in the proposed method, UAV-1 not only considers the total cost increases (the term before the plus sign $c_i(\Delta \mathbf{p}_i)$ in Equation (5), where the specific calculation is the term before the plus sign in Equation (8)) brought by task $T_2$ and task $T_3$, but it also considers the cost of returning to the position of the UAV (the term after the plus sign $c_i^o(\{j\})$ in Equation (5) and the specific calculation is the term after the plus sign in Equation (8)). These two factors together constitute the consideration $\eta_{12}^*$ of UAV-1 for task

$T_2$ and $\eta_{13}^*$ for task $T_3$, leading UAV-1 to select task $T_2$ with the minimum task consideration, which has no conflict with UAV-2. It can be found that the traditional method constantly adds outward tasks because the UAV blindly pursues the minimum cost increment and then competes with the distant UAV for tasks, while its nearby task $T_2$ cannot be completed due to the time limit. However, the proposed method not only completes all tasks but also reduces the task conflicts between UAVs, which is attributed to the proposed task consideration's ability to balance the task influence and task regression.

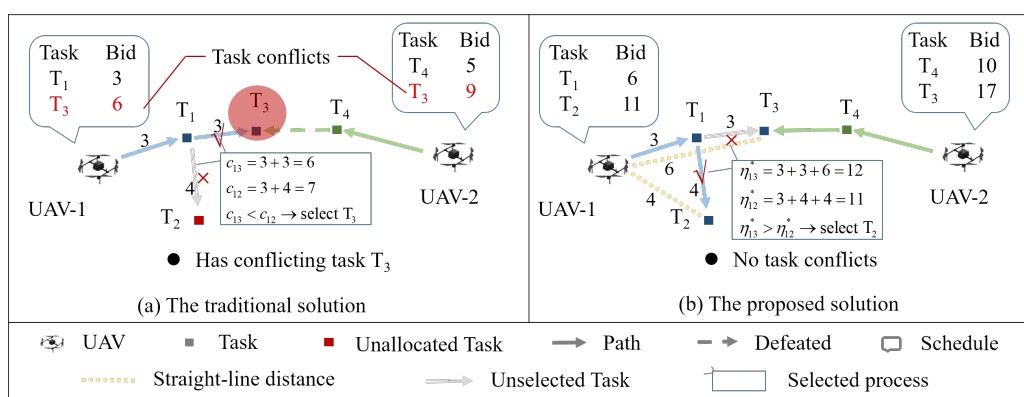

**Figure 1.** Comparison of two different methods to select tasks, where (**a**) is the traditional solution and (**b**) is the proposed solution.

The calculation method described above is that the task is already in the schedule, but when the task is not scheduled, the calculation results vary with the inserted locations of the task in the schedule. From the perspective of scheduling optimization, tasks should be inserted where they will have the least impact on the current schedule, that is, where the increase in path cost or time cost will be the smallest. Therefore, the task consideration of task $j$ to be added to UAV $i$'s schedule $\mathbf{p}_i$ is calculated as follows:

$$\eta_{ij}^*(\mathbf{p}_i \oplus \{j\}) = \min_{l \le |\mathbf{p}_i|+1} \{c_i(\mathbf{p}_i \oplus_l \{j\}) - c_i(\mathbf{p}_i)\} + c_i^o(\{j\}) \quad \forall j \in \mathcal{J}, \quad \forall i \in \mathcal{I} \qquad (6)$$

where $\eta_{ij}^*(\mathbf{p}_i \oplus \{j\})$ is the inclusion consideration of task $j$ for UAV $i$. Provided that task $j$ is added to the schedule $\mathbf{p}_i$, UAV $i$ will insert the task $j$ in the $l$-th location of schedule $\mathbf{p}_i$ that minimizes the difference value of the total scheduling cost.

Note that when task $j$ is not in the scheduling of UAV $i$, the task consideration of task $j$ is calculated as Equation (6), and when task $j$ is in the scheduling of UAV $i$, the calculation used is Equation (5). In fact, the results of these equations are the same and both the inclusion consideration and removal consideration are included in the task consideration.

The task consideration is compared among UAVs, and the task attribution is determined by the lowest task consideration so that the UAV swarm scheduling has better optimization. For example, the same task $j$, which is initially added to the schedule of UAV $i$, has a consideration of $\eta_{ij}$. Then, UAV $k$ wants to join task $j$, and its added consideration is $\eta_{kj}^*$. When $\eta_{ij}$ is greater than $\eta_{kj}^*$, that is to say, when the consideration of UAV $i$ to complete task $j$ is higher than that of UAV $k$, UAV $i$ will give up task $j$, and UAV $k$ will complete task $j$, which reduces the completion cost of task $j$ and optimizes the global scheduling. In other words, a task $j$ is executed by UAV $k$ when the following conditions are satisfied:

$$\eta_{ij}(\mathbf{p}_i \ominus \{j\}) > \eta_{kj}^*(\mathbf{p}_k \oplus \{j\}) \quad \forall j \in \mathcal{J}, \quad \forall i, k \in \mathcal{I} \qquad (7)$$

It can be found that when $i = k$, the removal and inclusion consideration are equal, and the attribution of task $j$ is also determined. When the attribution of all tasks is no longer transferred, the algorithm converges. In the process of practical application, each UAV $i$ in the swarm maintains a list of its own information about the global tasks, which mainly contains the winner $z_{ij}$ of task $j$ that local UAV $i$ believes and its winning bid $y_{ij}$. When

the task consideration $\eta_{ij}^*$ of local UAV $i$ for task $j$ is greater than the winning bid $y_{ij}$ of the current winner, UAV $i$ will not add task $j$ to its schedule, because the winner of current task $j$ is better. When the task consideration $\eta_{ij}^*$ of local UAV $i$ for task $j$ is less than the current winning bid $y_{ij}$, UAV $i$ will modify the winner of task $j$ to itself, $z_{ij} = i$, and the winning bid of task $j$ is set to its bid, $y_{ij} = \eta_{ij}^*$, and then inform the other UAVs through communication. If no other UAVs have different opinions, the attribution of task $j$ is determined.

### 3.2. Task Selection

After obtaining the information about all tasks, the swarm of UAVs first selects tasks independently, and the selection criteria will also affect the performance of swarm scheduling. As mentioned in the above section, the task consideration proposed in this paper has good exploratory and regressive properties, which can better optimize the scheduling. The specific calculation formula is as follows:

$$\eta_{ij}^*(\mathbf{p}_i\oplus\{j\}) = \min_{l\leq|\mathbf{p}_i|+1}\left\{c_{i,l}(\mathbf{p}_i\oplus_l\{j\}) + \sum_{q=l+1}^{|\mathbf{p}_i|+1}c_{i,q}(\mathbf{p}_i\oplus_l\{j\}) - \sum_{q=l}^{|\mathbf{p}_i|}c_{i,q}(\mathbf{p}_i)\right\} + \left\|\mathbf{l}_{io} - \mathbf{1}_{\{j\}}\right\|_2 \qquad (8)$$

where $\mathbf{p}_i\oplus_l\{j\}$ is the schedule after inserting task $j$ at the $l$-th location of schedule $\mathbf{p}_i$, $c_{i,q}(\mathbf{p}_i\oplus_l\{j\})$ is the cost of a task that is the $q$-th element in UAV $i$'s schedule $\mathbf{p}_i\oplus_l\{j\}$, and others are similar.

Since the UAV scheduling changes as tasks are added, the previously included tasks' considerations are calculated based on the previous scheduling. To ensure the optimization of the scheduling, the task consideration of the UAV needs to be updated after adding a new task. The calculation of the consideration of the newly added task is not based on the included tasks' consideration; it depends on the existing schedule and time, so it needs to be updated every time a task is added. The task consideration does not need to be updated in real time, and the consideration of all tasks in the current UAV schedule can be updated at one time after all tasks are added. In this case, the update of removal consideration (formula (5)) is used. Specifically, the consideration value of each task $j$ is calculated as follows:

$$\eta_{ij}(\mathbf{p}_i\ominus\{j\}) = c_{i,l}(\mathbf{p}_i) + \sum_{q=l+1}^{|\mathbf{p}_i|}c_{i,q}(\mathbf{p}_i) - \sum_{q=l}^{|\mathbf{p}_i|-1}c_{i,q}(\mathbf{p}_i\ominus\{j\}) + \left\|\mathbf{l}_{io} - \mathbf{1}_{\{j\}}\right\|_2 \qquad (9)$$

where $\mathbf{p}_i\ominus\{j\}$ is the schedule after removing task $j$ from UAV $i$'s schedule $\mathbf{p}_i$, $c_{i,q}(\mathbf{p}_i\ominus\{j\})$ is the cost of a task that is the $q$-th element in UAV $i$'s schedule $\mathbf{p}_i\ominus\{j\}$, and others are similar.

The process for each UAV to independently select the tasks can be briefly described in Algorithm 1. When the current schedule $\mathbf{p}_i$ of UAV $i$ has not reached its capacity $a_i$, UAV $i$ tries to select tasks to add to its schedule (lines 1–10). The general process is to calculate the inclusion consideration of the tasks that are not included (line 2) and then compare them with the current winning bid of tasks. If the maximal difference of winning bid $y_{i,j}$ and inclusion consideration $\eta_{i,j}^*$ is greater than 0 (line 3), select the task $j^*$ from which the difference between the winning bid and the task consideration is the largest (line 4), and obtain the best insertion location $l_{\{j^*\}}$ of task $j^*$ with maximum inclusion consideration $\eta_{i,j^*}^*(\mathbf{p}_i \oplus \{j^*\})$ (line 5). Then, task $j^*$ is inserted into schedule $\mathbf{p}_i$ at the best insertion location $l_{\{j^*\}}$ (line 6), the winner of task $j^*$ in the winner list $z_{ij^*}$ is updated as UAV $i$ and the winning bid is $\eta_{i,j}^*$ (line 7), and the expected execution time after task $j^*$ in schedule $\mathbf{p}_i$ is updated (line 8). Finally, the consideration of all tasks in UAV $i$'s schedule is updated at one time and correspondingly updated to the winning bid $\mathbf{y}_i$ (line 11).

---

**Algorithm 1** Task selection

---

1: **while** $|\mathbf{p}_i| \leq a_i$ **do**
2:      compute the list $\mathbf{J}_i^* \leftarrow \left[\eta_{i1}^*, \eta_{i2}^*, \ldots, \eta_{im}^*\right]$ by formula (8).
3:      **if** $\max_{j=1}^{m}\left\{y_{i,j} - \eta_{i,j}^*\right\} > 0$ **then**
4:          $j^* \leftarrow \arg\max_{j=1}^{m}\left\{y_{i,j} - \eta_{i,j}^*\right\}$
5:          $l_{\{j^*\}} \leftarrow \arg\eta_{i,j^*}^*(\mathbf{p}_i \oplus \{j^*\})$
6:          add task $j^*$ to scheduling $\mathbf{p}_i$ at location $l_{\{j^*\}}$
7:          update UAV $i$'s winner $z_{ij^*} = i$ and winnerbids $y_{ij^*} = \eta_{i,j^*}^*$.
8:          update the time $c_{i,j}(\mathbf{p}_i)$ of tasks after the task $j^*$ in scheduling $\mathbf{p}_i$
9:      **end if**
10: **end while**
11: update the consideration $\mathbf{J}_i$ of tasks in UAV $i$'s scheduling using Equation (9) and winnerbids $\mathbf{y}_i \leftarrow \mathbf{J}_i$.

---

*3.3. Swarm Consensus*

After the UAV selects the tasks to execute, it needs to communicate with the other UAVs to resolve the task conflict among UAVs. In order to reduce the amount of communication, this paper uses three vectors: task winners $\mathbf{z}_i$, task winning bids $\mathbf{y}_i$ and timestamps $\mathbf{t}_i$. By defining the conflict resolution rules that the UAVs abide by together, conflict-free swarm scheduling is obtained.

The conflict resolution rules mainly describe the process of how receiver $i$ handles the information received from sender $k$, which mainly includes the behaviors of updating, leaving, and resetting. The rules defined in this paper are similar to Reference [18], except for the following:

(1) In this paper, the task consideration, similar to the cost, is used as the bid, where the lower the better. In other words, the lower the bid, the better.
(2) Different from the traditional bid reset to 0, this paper resets to a maximum value.
(3) In order to improve the convergence speed, the receiver will update its information if the timestamp of the third party is equal.

Then, the UAV can obtain consistent global task attribution information, and then it needs to adjust the original schedule based on this consistent information, mainly for task removal. At present, the two most representative algorithms, CBBA and PI, adopt different removal methods. CBBA removes all tasks after the outbid task, which is exploratory but also results in more task conflict. The PI removes the outbid task with the maximum difference, updates the bid of remaining tasks, and re-compares it with the task winner. The updated conflict task with a lower bid than the winner's task is retained, which reduces exploration but also brings the disadvantage of insufficient optimization. Different from the traditional removal methods, this paper removes the outbid conflict tasks from the original schedule, which not only makes the task selection more exploratory but also ensures that the task attribution information is consistent with the conflict resolution.

Algorithm 2 outlines the process of swarm consensus. First, UAV $i$ communicates with all the UAVs $k$ in its neighborhood to resolve task conflicts and obtain consistent global task attribution information (line 1). Then, UAV $i$ removes all the conflicting tasks that are outbid by other UAVs and updates the time of the current schedule (lines 2–7). Finally, UAV $i$ updates the consideration of its current scheduled tasks and updates to the winning bid (line 8).

---

**Algorithm 2** Swarm consensus

---

1: UAV $i$ communicates with other UAV $k$ and gets the consensus of task winner list $\mathbf{z}_i$ and winning bid list $\mathbf{y}_i$
2: **for each** task $j$ in UAV $i$'s schedule $\mathbf{p}_i$ **do**
3:      **if** confict: $\mathbf{z}_{ij} \neq i$ **then**
4:          remove task $j$ from schedule $\mathbf{p}_i$
5:          update the time of current schedule
6:      **end if**
7: **end for**
8: update the consideration $\mathbf{J}_i$ of tasks in UAV $i$'s scheduling and winnerbids $\mathbf{y}_i \leftarrow \mathbf{J}_i$.

---

*3.4. Overall Process*

The overall flow of the proposed method is shown in Algorithm 3. Firstly, the UAV initializes the timer $T$ and the convergence flag $\omega$ (line 1). Then, when the convergence flag is false, the task selection of the UAV in Algorithm 1 and the swarm consensus in Algorithm 2 are performed in turn, where one cycle is one iteration. The convergence is checked at the end of each iteration. If there is no change in swarm scheduling, it is considered to have reached convergence. If it still does not converge, the timer is incremented by 1 and the loop proceeds to the next iteration until convergence occurs (lines 2–7).

---

**Algorithm 3** Overall Process

---

1: swarm of UAVs initializes timer $T \leftarrow 1$ and convergence flag $\omega \leftarrow 0$
2: **while** $\omega = 0$ **do**
3:      task selection as shown in Algorithm 1.
4:      swarm consensus as shown in Algorithm 2.
5:      $\omega \leftarrow checkConvergence$.
6:      $T \leftarrow T + 1$.
7: **end while**

---

In short, the UAV swarm scheduling can be derived through multiple iterations of the 'task selection and swarm consensus' phases. In the task selection phase, the proposed task consideration can reflect the impact of the task on the schedule and the regression of the task itself, so as to select the optimized task. The proposed new task removal method is used in the swarm consensus phase so that the UAV can have more task choices in the task selection phase and to increase the exploration of the schedule. The task consideration and the new removal method jointly improve the optimization of the algorithm.

**4. Experiment**

To verify the effectiveness of the proposed method, this paper is compared with the current two most representative algorithms, CBBA [18] and PI [19]. For convenience of comparison, the experiments in this paper are unified in the experimental scene of PI [19,21,24], in which there are two types of UAVs and tasks. The speed of the UAVs providing food is 50 m/s, and the task of aiding in serving the food lasts for 300 s. The speed of the medicine delivery UAV is 30 m/s an the medicine aid mission is completed in a duration of 350 s. All UAVs and tasks were randomly distributed in the 10 km $\times$ 10 km $\times$ 1 km area, and the deadline of tasks are randomly distributed within [0, 2000 s]. The Monte Carlo method was adopted to test 1000 situations with randomly distributed UAVs and tasks under the same $p$ and $n$ to ensure the objectivity of the experiment, where $p = m/n$ is the ratio of $m$ tasks and $n$ UAVs.

Figure 2 compares the task allocation number of the three methods, where it can be found that the method proposed in this paper allocates more tasks than the other two algorithms, enabling the UAV swarm to complete more tasks. When $p = 3$ and $n = 6$, the median number of allocated tasks for the proposed method is 17, with an increment of 6%

compared with CBBA and PI (both are 16). When $p = 5$ and $n = 16$, the median number of allocated tasks with CBBA is 63, with PI is 65, and the with proposed method is 66, representing 5% and 2% increments over CBBA and PI, respectively. The reason why the proposed task consideration (TC) method can increase the number of allocated tasks is that the task consideration considers the regression of the task so that the UAV can complete as many of the tasks in its vicinity as possible. Furthermore, the proposed new removal method in the swarm consensus phase enables the UAV to explore more tasks in the task selection phase. The two jointly improve the optimization of the proposed algorithm.

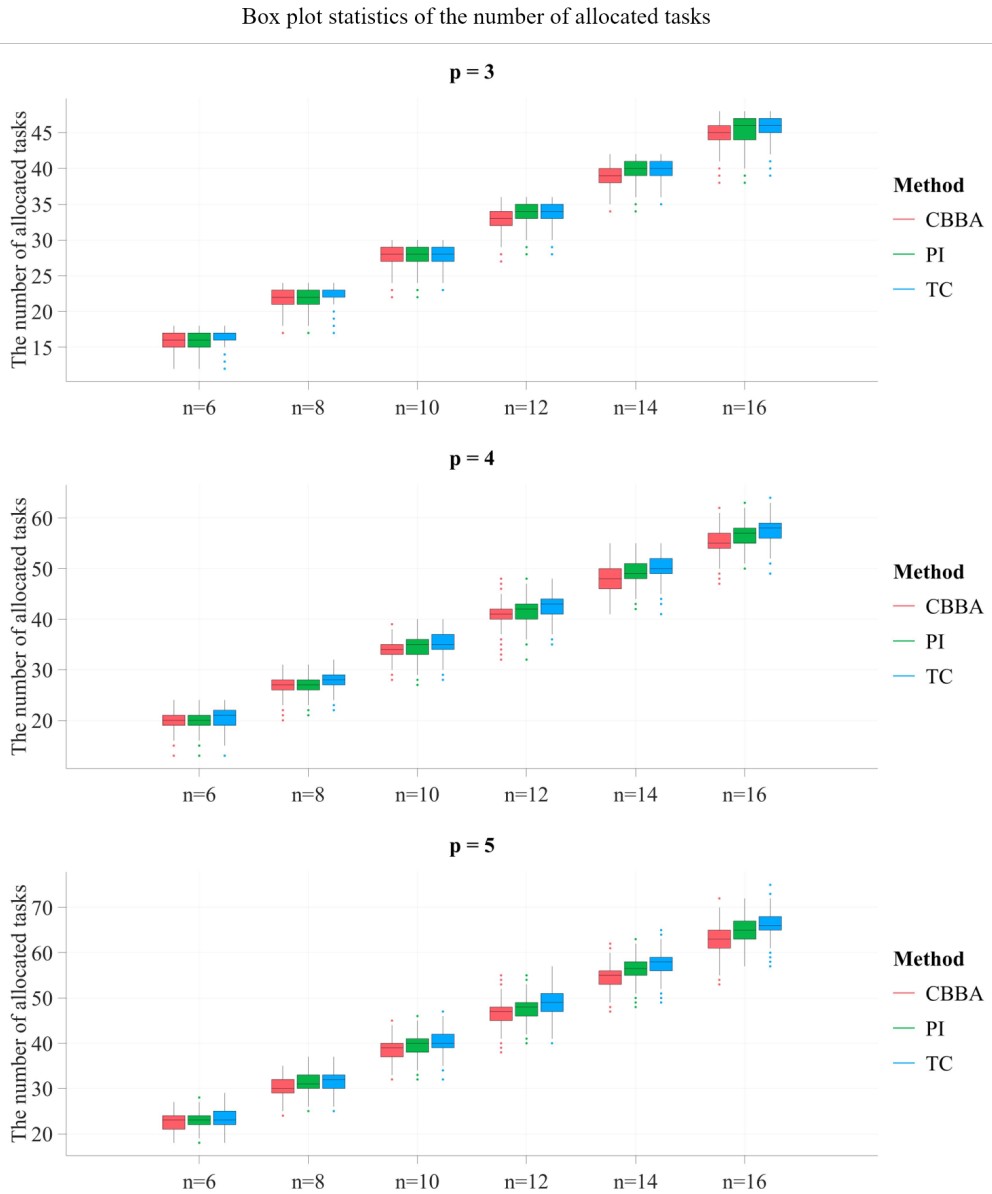

**Figure 2.** The comparison of three methods in terms of the number of allocated tasks.

Figure 2 shows that the proposed method can complete more tasks than the other two methods, and naturally, its completion time will be longer. In order to objectively compare the optimization of methods in terms of the total travel time, the comparison of the total travel time of different methods should be carried out under the same number of task allocations. Figures 3 and 4 respectively show the proportion and degree of reduction in the total travel time of swarm scheduling by the proposed method compared with the other two methods under the same number of task allocations, in which the color part is the comparison between the proposed method and CBBA method and the gray part is the

comparison with the PI method. It can be seen that the proposed method has a certain degree of improvement compared with the other two algorithms under all different $p$ and $n$. Compared with CBBA, the proposed method has the most improvement with 65–96% ameliorative solutions and a 2–6% average decrease in travel time. Compared with PI, the corresponding data are 31–48% and 1–4%, respectively. This effectively validates the optimization of the task consideration and task removal methods proposed in this paper, which balances the exploration and regression of tasks well.

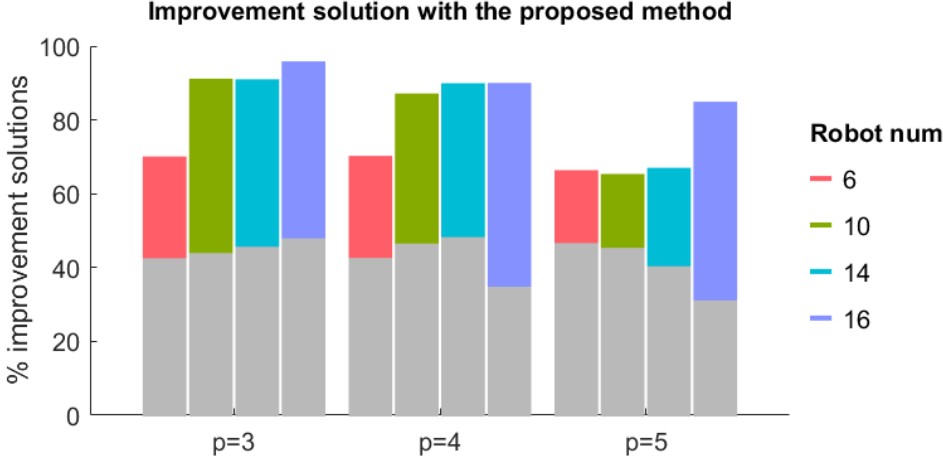

**Figure 3.** The percentage of improvement in solutions obtained with the proposed method over solutions obtained with the other two methods in the total travel time of swarm scheduling.

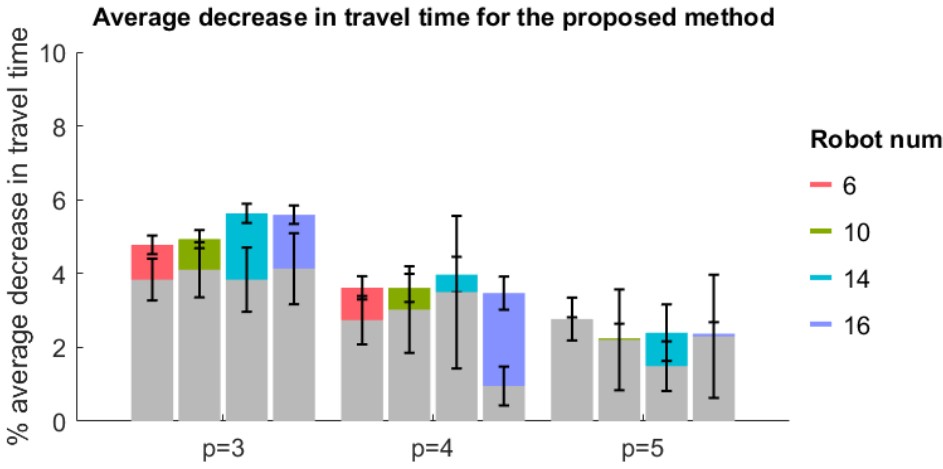

**Figure 4.** The average percentage increase and standard deviation in the total travel time of swarm scheduling over the other two methods.

Figure 5 shows the comparison of the three methods in terms of the number of iterations for various $p$ and $n$. It can be found that the advantages of the proposed method are more obvious when $p$ is larger. This is due to the regressive nature of the task consideration proposed in this paper, focusing on the exploration of tasks near the UAV and avoiding task conflicts with other UAVs caused by blind exploration, thereby reducing the number of iterations required for conflict resolution, which is especially obvious in the case of a large number of tasks per UAV. When $p$ is small, the proposed method has a small increase compared to CBBA because it needs to update the consideration to improve the optimality, but it is still far less than the number of iterations required by PI.

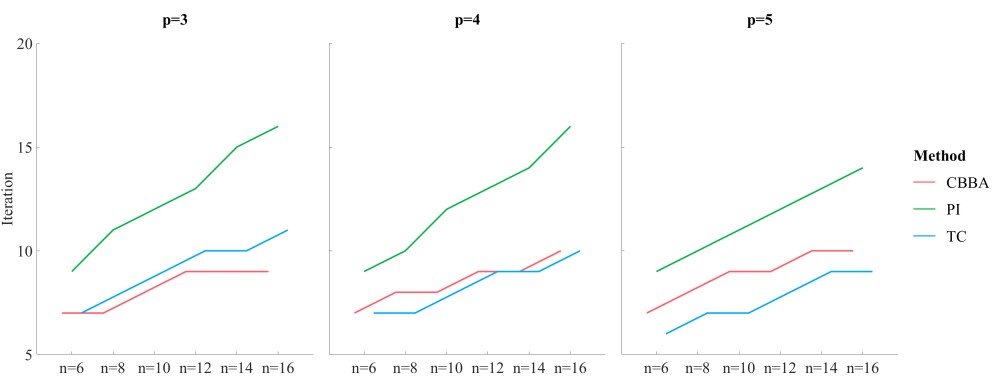

**Figure 5.** The iterations of the three methods to achieve convergence.

All in all, the proposed TC method has a certain improvement in optimization compared with the current two most representative algorithms, CBBA and PI, which is embodied in the number of task allocations and total travel time. Moreover, the number of iterations required by the TC method to achieve convergence is lower than the other two methods, which has a greater advantage when the number of tasks per UAV is large.

## 5. Conclusions

In this paper, a new concept called "task consideration" is introduced to improve the optimization and convergence of swarm scheduling. Task consideration balances the exploration of the UAV for the minimum impact task and the regression of neighboring tasks well. In addition, the modified conflict resolution rules are designed to achieve scheduling consensus, and a new removal method is proposed to further improve the optimization. Finally, through extensive Monte Carlo experiments, compared with CBBA and PI, the proposed method has greater task allocation and shorter travel times. With the increase in the average number of UAV tasks, the number of iterations is lower and the convergence is faster. In the future, we plan to research how to reduce the impact of the network on scheduling.

**Author Contributions:** Conceptualization, R.C. and J.L.; methodology, R.C.; software, R.C.; validation, J.L.; investigation, T.P.; writing—original draft preparation, R.C.; writing—review and editing, J.L. and T.P. All authors have read and agreed to the published version of the manuscript.

**Funding:** This research received no external funding.

**Institutional Review Board Statement:** Not applicable.

**Informed Consent Statement:** Not applicable.

**Data Availability Statement:** Not applicable.

**Conflicts of Interest:** The authors declare no conflict of interest.

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
