# Peer review of "Decentralized UAV Swarm Scheduling with Constrained Task Exploration Balance"

_drones, doi:10.3390/drones7040267_

Round 1
Reviewer 1 Report
The paper addresses a new scheduling technique for the members of a swarm and compares it to two widely referenced methods, CBBA and PI, using Monte Carlo simulations. Just like CBBA and PI, this method consists of a task selection and swarm consensus stage, but modifies the previous method bz not just looking outwards in the direction of the best cost benefit but also includes better exploration of adjacent tasks.
In general:
Style and grammar - the paper is filled with English grammar and style errors, which make the paper hard to read at times. It is essential to go through the paper thoroughly and address the English problems and make it easier to read.
Mathematics - there are various parameters in some of the equation that are not explained in the text. Sure, some of these are standard and, if you are familiar with the CBBA and PI or other scheduling work known, but for completeness they should be included. For example: I and J in equations (2) and (3). Furthermore, make sure the indexing is consistent throughout the paper. Individual issues will be addressed below.
Abstract - the first part of the abstract should be rewritten so it more clearly states how this method approves over previous methods such as CBBA and PI.
Page 3: In line 118 Figure 1 is referenced to explain the what is new in their method. It would really help if this example would be directly related to the mathematics introduced in eqn (5) specifically, and latter related to the specific implementation equations ((8) and (9) as well as algorithm 1. This would make the explanation of the algorithm more intuitiv.
Page 4: in equation (8) the '[p]+1 is above the min, this is a different notation that used in equation (6). Also, in the text it is important to relate this equation to equation (6) better.
Page 5: in equation 9, the authors use a 'k' on the right-hand side, but this 'k' does not show up on the left-hand side indicated that is was used through an operator like done in equation (8).
Page 5: also, the importance of equation (9) and (8) is obvious from equation (7), but how is equation (9) used in algorithm 1 on page 5? Explain.
Page 5: in line (5) the variable for the winning bid (y_i,k) is introduced as it is logically part of algorithm 1 (like in CBBA and PI). However, this is the first time the list of winning bids is described. It is important to explain what this is so in-depth knowledge of auction-based algorithm (through the references to CBBA and PI) is not required.
Page 5: check line 5 of algorithms 1 for correctness.
Page 7: here the authors describe the experiment that was also used to evaluate the I algorithms. A little more description of the algorithm as well as the costs function may be include here.
Page 7: the plot are a bit small, it is recommended to increase the plot size and also add gridlines.
Page 9: in the caption of Figure 5 provide some more detail, like "The iteration of the three methods to achieve converge" or something.
Again, the authors really should go through the paper and improve the grammar and style of the English. The paper is really hard to read.
Reviewer 2 Report
In the paper, a new concept of Task Consideration was proposed to improve scheduling optimization and convergence. Rules of conflict resolution were modified to adapt to the method, and the new removal algorithm was used to increase the exploration of tasks. The paper is innovative, but it has some problems:
1. In the abstract, the results section is not clearly mentioned.
2. The English language level in this paper is not up to the standards that are required for publication in an international research journal. There are so many spelling mistakes. Authors are advised to perform a major revision in English.
3. The comparison algorithm used in the simulation experiment is quite old, and it should be compared with some other algorithms of relevant literatures in the past five years.
4. Please highlight how the work advances or increments the field from the present state of knowledge and provide a clear justification for your work. Authors need to provide critical comments on the contribution and shortcomings.
5. Most references should be selected from the latest ones and categorized and elaborated to show their advantages and disadvantages.
Reviewer 3 Report
This research addressed an interesting topic related to decentralized UAV scheduling.
Some comments could be considered.
- Explore further studies and provide a comparison related to your work, e.g., the strongest points of your work, the contributions and limitations of the previous work concerning your contribution, something like a summarized table.
-
- Simplification the equation if it is possible?
- Minimize the abbreviations, and please add a table for them to make the paper easier and increase its readability.
- Also, it will be helpful if you provide a summary at the end of each section to conclude your work in the reader's mind.
- Further discussion of the results
-
- Also, You can use and cite the following references, which are related to your work regarding computing and scheduling using emerged technologies blockchain, IoT and 6G
Computing in the sky: A survey on intelligent ubiquitous computing for uav-assisted 6g networks and industry 4.0/5.0
Green internet of things using UAVs in B5G networks: A review of applications and strategies
Round 2
Reviewer 2 Report
1. The paper should emphasize contributions in the conclusion section.
2. The references need to be modified as required, such as Ref. 20.
